# Making Self-supervised Learning Robust to Spurious Correlation via Learning-speed Aware Sampling

**Weicheng Zhu**                                                    *jackzhu@nyu.edu*
*New York University*

**Sheng Liu**                                                        *shengl@stanford.edu*
*Stanford University*

**Carlos Fernandez-Granda**                                          *cfgranda@cims.nyu.edu*
*New York University*

**Narges Razavian**                                         *Narges.Razavian@nyulangone.org*
*NYU Grossman School of Medicine*

**Reviewed on OpenReview:** *https: // openreview. net/ forum? id= 8mgX3Uw2Ea*

## Abstract

Self-supervised learning (SSL) has emerged as a powerful technique for learning rich representations from unlabeled data. The data representations can capture many underlying attributes of data, and are useful in downstream prediction tasks. In real-world settings, spurious correlations between some attributes (e.g. race, gender and age) and labels for downstream tasks often exist, e.g. disease findings are usually more prevalent among elderly patients. In this paper, we investigate SSL in the presence of spurious correlations and show that the SSL training loss can be minimized by capturing only a subset of conspicuous features relevant to those sensitive attributes, despite the presence of other important predictive features for the downstream tasks. To address this issue, we investigate the learning dynamics of SSL and observe that the learning is slower for samples that conflict with such correlations (e.g. elder patients without diseases). Motivated by these findings, we propose a learning-speed aware SSL (LA-SSL) approach, in which we sample each training data with a probability that is inversely related to its learning speed. We evaluate LA-SSL on three datasets that exhibit spurious correlations between different attributes, demonstrating the enhanced robustness of pretrained representations on downstream classification tasks.

## 1 Introduction

Self-supervised learning (SSL) which learns data representations without explicit supervision has become a popular approach in various vision tasks (Chen et al., 2020; He et al., 2020; Caron et al., 2020; Grill et al., 2020; Zbontar et al., 2021; Caron et al., 2021). The learned representations are often used for various downstream tasks through supervised fine-tuning with task-related labeled data. Despite the overall effectiveness of SSL methods for many downstream tasks, it is crucial to make sure representations learned by SSL are not biased. Understanding and addressing the potential sources of bias is essential for ensuring the robustness and reliability of SSL methods in practice.

One source of bias stems from spurious correlations, where some attributes of the data are correlated to the target labels due to the imbalanced data distribution rather than causal relationships. When spurious correlation is present, neural networks may pick up the correlation as a shortcut, and use features corresponding to spurious attributes to achieve good overall performance on the task (Nam et al., 2020). For instance, the prevalence of pneumonia in different hospitals being different does not imply that patients in some

hospitals have a higher risk of pneumonia, but neural networks may make predictions based on hospital-related features in the radiography (Zech et al., 2018). Spurious correlations can therefore negatively impact out-of-domain generalization and fairness in real-world applications, especially in healthcare (Liu et al., 2020; Seyyed-Kalantari et al., 2021; Nauta et al., 2021; Zhu et al., 2022). In principle, one would expect SSL pretraining to be robust to spurious correlations, since task labels are not directly involved in training and the SSL pre-trained representation could potentially learn diverse features from the data. However, here we show that SSL may still introduce biases to representations when the data distribution depends strongly on spurious attributes.

Feature suppression is a common phenomenon in SSL, where the model prioritizes learning easy-to-learn features as shortcuts, while neglecting features related to other attributes (Robinson et al., 2021). We observe that when some attributes are correlated with one or several other attributes, the feature suppression nature of SSL can result in less discriminative representations for some attributes while being more discriminative for others. This bias in the learned representation can hinder the performance of downstream tasks that rely on those attributes which the learned representations are less discriminative to. Therefore, understanding and mitigating the biases in SSL is crucial for developing accurate and fair SSL pretrained models.

In this work, we aim to address an understudied problem of SSL: *how to improve the robustness of SSL to spurious correlations among underlying attributes?* Previous studies show that sampling conditioning on some predefined attributes in SSL can force the data representations to be less discriminative to them (Tsai et al., 2022; Ma et al., 2022). This can potentially mitigate the spurious correlation when leveraging spurious attributes as conditions. However, pretraining with SSL typically only utilizes unlabeled data (i.e. the images) and does not leverage any extra labels, therefore potential attributes may not be known in advance during training. As an alternative, we introduce the learning speed as a proxy for spurious attributes. We find that the samples whose values do not follow such correlation with the spurious attributes are often learned slower in SSL than the samples that align with the correlation. Based on these observations, we propose a learning-speed aware sampling method for SSL (LA-SSL). We evaluate the learning speed of the feature extractor on each example during training and sample each example with a probability that is negatively related to its learning speed. This forces the SSL to learn more discriminative features from examples that do not follow the spurious correlation, which helps the model learn representations containing rich and diverse features, improving its generalization on a variety of downstream tasks. In summary, our contributions are the following:

- We investigate the training behavior of SSL methods on datasets with spurious correlations and observe that there exists a difference in learning speed between examples aligned and conflicting with such correlations;

- We propose a novel SSL method, LA-SSL, where we dynamically sample training samples based on their learning speed. This reduces the impact of spurious correlations in the dataset by upsampling the examples that may conflict with spurious correlations.

- We show that LA-SSL improves the robustness of previous SSL frameworks in terms of the linear probing accuracy in downstream classification tasks on three datasets where spurious correlations among different attributes are present.

## 2 Related Work

**Self-supervised learning**    Self-supervised learning (SSL) is widely used to learn the representations for large unlabeled image datasets. A popular method, contrastive learning, seeks to learn representations by contrasting positive and negative pairs of samples (Chen et al., 2020; He et al., 2020; Caron et al., 2020). Other non-contrastive methods leverage similarities, cross-correlation, variance, or covariance in the representation space (Grill et al., 2020; Zbontar et al., 2021; Caron et al., 2021; Bardes et al., 2021), to learn representations such that two augmentations from the same image are close while avoiding mode collapse. Other works learn the data representation by teaching the network to reconstruct the corrupted images (Bao et al., 2022; He et al., 2021).

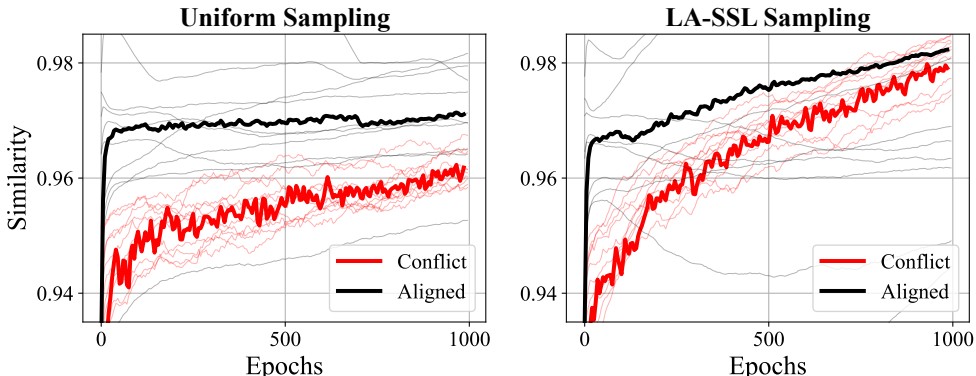

Figure 1: The similarity between the representations of two augmented views on a benchmark dataset (95% correlation-aligned corrupted CIFAR-10) during training. The thick curves represent the mean of similarities of correlation-conflicting and -aligned examples, while the light curves represent the similarity mean of each class. The comparison highlights the network's faster learning on aligned examples compared to conflict examples. The reweighted sampling approach in LA-SSL narrows the gap between conflict and aligned examples by upsampling the examples that learn slower. (Appendix B.1 shows additional examples).

**Spurious correlation**  Several supervised learning methods, such as GroupDRO (Sagawa et al., 2019), LfF (Nam et al., 2020), JTT (Liu et al., 2021), SSA (Nam et al., 2022), and LC (Liu et al., 2023a), address the issue of model bias caused by spurious relationships in the labeled data. However, these methods require either strong prior knowledge about the confounding attributes involved in the spurious correlation (e.g., GroupDRO), or rely on the assumption that there exists an unknown attribute correlated with the target label of a specific task. In this paper, we focus on learning robust representations from unlabeled datasets with self-supervised methods, which is distinct from the de-biasing algorithms in the supervised learning setting. While the DFR method demonstrates the ability to mitigate spurious correlation by only fine-tuning the last layer on top of biased data representations (Kirichenko et al., 2022; Izmailov et al., 2022), the desire for robust representations in SSL remains crucial for generalization across a wide range of downstream tasks without relying on assumptions of spurious correlation.

**De-biased self-supervised pretraining**  Sampling bias is a source of bias in self-supervised learning (Arora et al., 2019; Chuang et al., 2020; Assran et al., 2022). This bias impedes the network's ability to learn accurate representations of minority subgroups, which are common in the dataset impacted by spurious correlation. To mitigate this bias, researchers have proposed conditional sampling techniques, which improve fairness among subgroups (Ma et al., 2022; Tsai et al., 2022). However, these techniques rely on group information, which is typically unavailable in large unlabeled datasets, contradicting the goal of SSL. Similarly, the unsupervised domain generalization frameworks are constrained by the requirements of the domain information (Zhang et al., 2022; Liu et al., 2023b). Meanwhile, various alternative SSL methods have been introduced to overcome this challenge and improve data representations by avoiding shortcuts. These methods include incorporating hard negatives (Robinson et al., 2020), adversarially perturbing data representations (Robinson et al., 2021), and employing network pruning or masking techniques (Lin et al., 2022; Hamidieh et al., 2022) during training. However, these methods do not specifically address the data imbalance caused by spurious correlation. While network pruning has been utilized for pretraining on a biased dataset (Hamidieh et al., 2022), it is only used in downstream tasks on group-balanced datasets. In contrast, we focus on analyzing and improving the robustness of SSL pretraining for the downstream tasks where spurious correlation persists.

## 3 Preliminary Settings: Spurious Correlation in Self-supervised Learning

A key assumption in the success of self-supervised learning (SSL) is that minimizing its loss function enables the network to learn representations that are discriminative to the underlying attributes of the data. However, SSL fails to achieve this when certain attributes in the training samples are strongly correlated.

We assume that the training data $\mathcal{D}$ has underlying attributes $Z_1, \cdots, Z_p$, where each attribute $Z_i$ can be discretized into $K_i$ categories and follows a distribution $p_{Z_i}$. By combining the values $(z_i, z_j) \in Z_i \times Z_j$, any two attributes $Z_i, Z_j$ can form $K_i \times K_j$ subgroups. When there is a strong correlation between $Z_i$ and $Z_j$, the training samples will not be evenly distributed among the subgroups. We refer to the samples in subgroups with high joint probabilities $p_{Z_i, Z_j}(z_i, z_j)$ as correlation-*aligned* samples, while those with low joint probabilities are referred to as correlation-*conflicting* samples. For instance, in medical datasets, the occurrence of diseases is usually positively correlated with age. Hence, old sick and young healthy patients are often correlation-aligned samples, while young sick and old healthy patients are correlation-conflicting samples.

The bias in SSL arises primarily from the correlation-aligned samples. Within these correlation-aligned examples, the SSL loss can be minimized by only learning representations discriminative to some easy-to-learn feature $Z_i$ while suppressing features for $Z_j$ (Robinson et al., 2021). This bias is reinforced when optimizing the loss function via randomly sampled training data, as the correlation-aligned samples constitute the majority of the training set. Although correlation-conflicting samples, which contain diverse combinations of attribute values for $Z_i$ and $Z_j$, could potentially guide the model to be discriminative towards $Z_j$, their scarcity prevents them from mitigating this bias effectively. In such situations, the learned representations may succeed in classifying labels relying on $Z_i$ in downstream tasks, but they may fail to classify labels relying on $Z_j$ accurately. For instance, in medical datasets, if $Z_i$ represents patient age and $Z_j$ represents the presence of a specific disease, the model might effectively classify age groups but struggle to detect the disease in downstream tasks.

To better understand the gap in the discriminability of the learned representations across attributes, we conduct spectral analysis in Section 5.5, which quantifies the strength of dominating features learned by SSL. It indicates that a few large singular values can greatly weaken the informative signals of eigenvectors corresponding to smaller singular values (Chen et al., 2019), suppressing the discriminability of learned representations for subdominant attributes like $Z_j$.

## 4 Learning-speed Aware Sampling

In this section, we present a learning-speed aware sampling schema in self-supervised learning (LA-SSL) to improve the robustness of representations to spurious correlations. We first introduce conditional sampling, which facilitates the learning of a fair representation based on known attributes in Section 4.1. Then we leverage observations on learning-speed differences to stratify examples that either align or conflict with the spurious correlation in Section 4.2. Based on these two insights, LA-SSL tracks the learning speed of each example during training as the proxy of correlation-aligned or conflicting subgroups. LA-SSL incorporates conditional sampling based on learning speed, assigning each example a probability that is inversely related to its learning speed (see Section 4.3).

Most recent SSL approaches (Chen et al., 2020; He et al., 2020; Grill et al., 2020; Zbontar et al., 2021; Caron et al., 2021) adopt a two-branch structure, where two views of an example obtained through random augmentations are encoded, and the representations of them are encouraged to be similar. In this section, we mainly present the application of LA-SSL on top of one of these frameworks, SimCLR (Chen et al., 2020). Then we also discuss the applicability of our method to other two-branch SSL frameworks and generative frameworks with only one branch in Section 5.4.

### 4.1 Conditional sampling in SSL

Conditional contrastive learning proves to be effective in learning fairer representations with respect to specific attributes by excluding information on these attributes in the representations (Ma et al., 2022; Tsai et al., 2022). This approach can therefore be employed in datasets with spurious correlations to mitigate the influence of confounding attributes. The key idea is to minimize the InfoNCE loss on data pairs sampled from all the training data that share the same value in an attribute $Z$, denoted as $\mathcal{D} \mid Z$. Formally, it minimizes

the loss function in Equation 1:

$$\mathcal{L}_{\text{C-SSL}} = \mathbb{E}_Z \left[ \underset{\{x_i\}_{i=1}^b \sim \mathcal{D}|Z}{\mathbb{E}} \left[ -\log \frac{sim(x_1^{\text{aug 1}}, x_1^{\text{aug 2}})}{sim(x_1^{\text{aug 1}}, x_1^{\text{aug 2}}) + \sum_{i=2}^b sim(x_1^{\text{aug 1}}, x_i)} \right] \right] \tag{1}$$

The similarity score $sim(\cdot, \cdot) : \mathbb{R}^m \times \mathbb{R}^m \to \mathbb{R}$ is defined as $sim(x, x') = \exp(f_\psi(x) \cdot f_\psi(x')/\tau)$ for any $x, x' \in \mathbb{R}^m$, where $f_\psi = \psi \circ f$, in which $f : \mathbb{R}^m \to \mathbb{R}^d$ is the feature extractor mapping the input data to a representation, $\psi : \mathbb{R}^d \to \mathbb{R}^{d'}$ is a projection head with a feed-forward network and $\ell_2$ normalization, and $\tau$ is a temperature hyperparameter. The first expectation is taken over $z$ uniformly drawn from all possible values of attribute $Z$; the second expectation is taken over samples $x \in \mathbb{R}^m$ drawn uniformly from the subset of the training set with attribute $Z = z$. Minimizing the loss brings the representation of two random augmentations $x^{\text{aug 1}}, x^{\text{aug 2}}$ on an instance $x$ closer, and pushes the representation of $x$ away from representations of $b - 1$ other examples $\{x_i\}_{i=2}^b$ in the training set.

While conditional contrastive learning is able to learn a fairer representation across various subgroups, it is not feasible in the typical SSL practice as it requires additional attributes. However, we only have access to unlabeled image data for SSL pretraining. Therefore, we propose an alternative approach to identify the subgroups of attribute values in spurious correlation.

### 4.2 Learning speed difference in SSL

Previous studies show that different training samples are learned at different paces in the supervised learning (Sagawa et al., 2019; Nam et al., 2020). When using empirical risk minimization (ERM) with cross-entropy loss to train a classifier, if the labels are biased due to a spurious relationship between some confounding attributes and the labels, the training loss of correlation-aligned samples decreases at a faster rate compared to correlation-conflicting samples. Note that variables used as labels here can be viewed as one attribute of data in the self-supervised setting. This phenomenon implies that the network can quickly overfit to these biases, using them as shortcuts. Since the features associated with these confounding attributes are easier to learn than the original task, the neural network tends to memorize the labels that align with the spurious relationship first.

We observe a similar phenomenon in SSL pretraining even in the absence of labeled supervision. We introduce the concept of *learning speed* for a sample, defined by a function $s : \mathbb{R}^m \to \mathbb{R}$ that maps the training example to a score. In SSL frameworks with two branches, the network is trained to enforce the representations of two randomly augmented versions of the same sample to be closer. Consequently, it is natural to evaluate the learning speed of a training example $x$ by the similarity between the representations of its two augmentations, such that $s(x) = sim(x^{\text{aug 1}}, x^{\text{aug 2}})$.

Figure 1 shows the training dynamics of SimCLR and LA-SSL on a benchmark dataset of spurious correlation, corrupted CIFAR-10 (Hendrycks & Dietterich, 2019), which simulates a spurious relationship by associating corruption type with the object labels. In this illustration, the training set is partitioned into two groups: the correlation-aligned group, whose target labels are perfectly correlated with the corruption types, and the correlation-conflicting group, whose target labels are uncorrelated. The plot shows that the feature extractor in SimCLR learns faster on the correlation-aligned examples than on the correlation-conflicting examples. Similar to supervised learning, SSL also tends to capture the easily learnable attributes first. Motivated by this observation, we propose a method to leverage the imbalanced learning speed to enhance the quality of representations learned through SSL.

### 4.3 Conditional Sampling based on Learning Speed

Since the learning speed varies with subgroups formed by the values of attributes involved in spurious correlation, we use them as the proxy of those attributes. Instead of sampling conditioning on the values of attributes directly, we sample training data based on their learning speed. The goal is to encourage the model to learn faster for the correlation-conflicting samples in the spurious relationship. This is achieved by dynamically adjusting the probability of sampling each training data.

---

**Algorithm 1** Learning-speed Aware Self-supervised Learning (LA-SSL)

---

**Require:** Samples $x_1, \cdots, x_n \in \mathcal{D}$;
**Require:** Scaling function $h$; smoothing constant $\eta$;
1:   *# Initialize sampling probability with equal weights*
2: $\pi \leftarrow (\pi_1, \cdots, \pi_n)$, where $\pi_i = \frac{1}{n}, \forall i$
3: **for** $t = 1$ to T **do**
4:      *# Compute the similarity between two views*
5:      $s_i^{(t)} = sim\left(x_i^{\text{aug}1}, x_i^{\text{aug}2}\right), \forall i$
6:      *# Smooth the similarities by EMA*
7:      $s_i^{(t)} \leftarrow (1-\eta)s_i^{(t-1)} + \eta s_i^{(t)}, \forall i$
8:      *# Update the probabilities after warmup epochs*
9:      **if** $t > T_{\text{warmup}}$ **then**
10:          *# Compute weights from the similarity scores*
11:          $\pi_i \leftarrow h(s_i^{(t)})/\sum_{i=1}^n h(s_i^{(t)})$
12:     **end if**
13:     *# Optimize the LA-SSL loss via Eq.(2)*
14:     $f_\psi^{(t+1)} \leftarrow \text{argmin}_{f_\psi} \mathcal{L}_{\text{LA-SSL}}(f_\psi^{(t)})$
15: **end for**

---

Let $\Pi$ denote a categorical random variable on sample space $\{1, \cdots, n\}$ and $\pi \in \mathbb{R}^n$ denote the probability of sampling each training example. During training, we sample the indices of examples from $\Pi$ and minimize the InfoNCE loss on training data with corresponding indices. The loss function of LA-SSL is defined as:

$$\mathcal{L}_{\text{LA-SSL}} = \mathop{\mathbb{E}}_{\{k_i\}_{i=1}^b \sim \Pi} \left[ -\log \frac{sim(x_{k_1}^{\text{aug}1}, x_{k_1}^{\text{aug}2})}{sim(x_{k_1}^{\text{aug}1}, x_{k_1}^{\text{aug}2}) + \sum_{i=2}^b sim(x_{k_1}^{\text{aug}1}, x_{k_i})} \right] \tag{2}$$

In LA-SSL, we aim to upsample the training data with lower learning speed, because the correlation-conflict samples are usually learned slower. Therefore, for each training sample $x_i$, we dynamically update $\pi_i$ with a weight that is inversely related to the learning speed $s_i$ as described in Algorithm 1. The algorithm is computationally and memory-efficient, requiring only the additional resources needed to compute and store similarity scores for each training sample, without introducing extra forward or backward propagation steps.

To monitor the learning speed for each training example $x_i$, we compute the similarity between the representations of its two randomly augmented versions at each epoch. Since the similarity score can be affected by randomness in augmentation strengths, we use exponential moving average (EMA) on $s(x_i)$ across previous training epochs to compute a stabilized learning speed $s_i$ for training example $x_i$'s. Then we compute the weights inversely related to the learning speed. The inverse relationship is set by a linear scaling function defined as:

$$h(s_i) = [s^* - \gamma(s_i - s^*)]^+ \tag{3}$$

where $s^* \in \mathbb{R}$ is a threshold selected as the $r$-percentile among the EMAs of learning-speed $s_i$'s from all training data, and $\gamma \in \mathbb{R}$ is a constant that increases the margin between examples with slow and fast learning speed. $r$ and $\gamma$ are hyperparameters. We typically choose small values of $r$ and $\gamma$ greater than 1 to differentiate the underrepresented samples in spurious relationships with lower learning speeds. The choice of hyperparameter is analyzed by sensitivity analysis in Appendix B.4. The probability $\pi$ is then computed by the weights of inversely scaled learning speed for each training example: $\pi_i = h(s_i)/\sum_{i=1}^n h(s_i)$. We only update the weights after the warm-up epochs to ensure that the network has already learned some features instead of being random. As demonstrated in the right panel of Figure 1, this sampling scheme indeed results in a better-synchronized learning speed between correlation-aligned and -conflicting examples.

## 5   Experiments

We introduce three datasets used to evaluate the robustness of SSL to spurious correlation in Section 5.1. We show that LA-SSL consistently improves the performance of SimCLR on the downstream tasks impacted by

Table 1: The classification accuracy (%) evaluated on group balanced test sets of corrupted CIFAR-10 with varying correlation-aligned ratio (k%). The strength of spurious correlation grows with $k$. LA-SSL outperforms the uniform sampling baseline in all scenarios.

| Method | Correlation-aligned proportion ($k\%$) | | | |
|--------|------|------|------|-------|
|        | 95%  | 98%  | 99%  | 99.5% |
| SimCLR | 44.08 | 36.60 | 29.74 | 22.99 |
| LA-SSL | **48.02** | **40.49** | **31.55** | **24.17** |

spurious correlations in the three datasets in Section 5.2 and particularly, under multiple spurious correlations in Section 5.3. In Section 5.4, we show that LA-SSL can be generalized to other two-views and generative-based SSL methods. In Section 5.5, we explain the robustness of LA-SSL pretrained representations from the perspective of spectral decomposition.

## 5.1 Datasets and metrics

We evaluate the proposed framework on three datasets that have inherent spurious correlations. Since SSL methods require a large amount of data to train the data representation, we experiment on datasets with large sample sizes instead of other popular benchmark datasets for spurious correlation problems with limited samples, like Waterbirds (Welinder et al., 2010). More details about the dataset and experiment settings are provided in Appendix A.

**Corrupted CIFAR-10** (Hendrycks & Dietterich, 2019) is a synthetic dataset generated by corrupting 60,000 images in CIFAR-10 with different types of noises. To simulate the spurious correlation between label and noise type, $k\%$ images are corrupted with a fixed noise type corresponding to their labels, while the other $1 - k\%$ of images in each class are corrupted by a random noise type. The higher the value of $k$ is, the stronger the correlation between noise types and labels. The downstream task we evaluate is object classification. Note that we avoid using augmentation that is similar to corruption during SSL such as Gaussian blur.

**CelebA** (Liu et al., 2015) is a dataset that contains 202,599 images of celebrities along with various attributes associated with the images. One attribute indicates whether the person has *Blond Hair*, which exhibits a strong spurious correlation with the *Male* attribute. Only 2% of males in the dataset have blond hair as opposed to 24% in females. The downstream task we evaluate is hair color classification.

**MIMIC-CXR** (Johnson et al., 2019) is a dataset of 377,110 chest X-ray images labeled with the diagnosis and demographic information of patients. There is a spurious correlation between the *No Findings* attribute and the *Age* attribute. The *No Findings* attribute is more common among younger patients in the dataset, while older patients are more likely to have some diseases. We stratify patients into two sub-cohorts by whether they are younger or older than 90. We also create a *synthetic version* of MIMIC-CXR where we intentionally simulate a strong gender/disease bias by downsampling the no-finding male patients.

**Evaluation metrics** We evaluate the performance of standard (randomly sampled) SSL and LA-SSL on the test set of these three datasets. The test set of corrupted CIFAR-10 is balanced among 10 classes, and the images in each class are randomly corrupted. Therefore, we report the accuracy of classification on the test set. The test sets of CelebA and MIMIC-CXR are still highly imbalanced, so we evaluate the precision and recall together with AUROC on the subgroup that performs worse. For the MIMIC-CXR task, we evaluate no-finding classification as the downstream task.

## 5.2 Results

In this section, we compare the performance of LA-SSL with uniform random sampling based on a common SSL framework, SimCLR (Chen et al., 2020) with Resnet-50. Both SSL framework and linear classifier for downstream tasks are trained on the same training dataset. Table 1 reports the held-out test set accuracy of models trained on corrupted CIFAR-10 with varying levels of spurious correlations between the type of

Table 2: Downstream task performance on the underperforming subgroups affected by the spurious correlation in CelebA and MIMIC-CXR. LA-SSL demonstrates improved precision and recall on those imbalanced subgroups in both datasets. The comprehensive performance of all the subgroups is shown in Appendix B.3.

| Dataset | Method | AUC | Precision | Recall |
|---------|--------|-----|-----------|--------|
| CelebA (Gener-Male) | SimCLR | 95.14 | 63.80 | 37.22 |
|  | LA-SSL | **95.63** | **66.66** | **38.88** |
| MIMIC-CXR (Age-Old) | SimCLR | 73.75 | 37.40 | 31.61 |
|  | LA-SSL | **74.89** | **40.00** | **34.83** |

Table 3: Performance on the synthetic MIMIC-CXR dataset simulating spurious correlation with gender and age attributes. LA-SSL improves subgroup accuracy under both spurious correlations.

| Attribute | Method | AUC | Precision | Recall |
|-----------|--------|-----|-----------|--------|
| **Gender**-Male | SimCLR | 76.39 | 55.95 | 56.13 |
|  | LA-SSL | **77.56** | **59.53** | **57.55** |
| **Age**-Old | SimCLR | 69.28 | 32.70 | 61.47 |
|  | LA-SSL | **70.68** | **34.71** | **61.67** |

corruption and the target label. The spurious correlation is only present in the training/validation set and the test does not have any spurious correlation. Both SSL methods exhibit lower performance as the proportion of correlation-aligned samples increases, indicating their susceptibility to spurious correlation. However, LA-SSL demonstrates its capability to learn a more robust representation for downstream tasks by achieving a relative improvement of 8.93%, 10.63%, 6.07%, and 5.13% at each corruption level, respectively. Additionally, Appendix B.5 reports comparisons between LA-SSL and an robust SSL method to avoid shortcut learning by adversarially perturbing data representations (Robinson et al., 2021), demonstrating LA-SSL still outperforms the robust SSL in the presence of spurious correlation. Notably, LA-SSL pretraining, coupled with simple linear probing on frozen representations, achieves even better performance than existing supervised de-biasing algorithms for spurious correlations, such as JTT (Liu et al., 2021) whose accuracy on corrupted CIFAR-10 is 24.73%, 26.90%, 33.40% and 42.20%, respectively. This shows the robustness of LA-SSL to spurious correlations.

Table 2 reports the performance on two real-world datasets CelebA and MIMIC-CXR, which are both affected by spurious correlations. The male subgroup in CelebA and the old subgroup in MIMIC-CXR have inferior performance compared to the general population. LA-SSL consistently improves precision and recall in these underrepresented subgroups. This further confirms its effectiveness in addressing spurious correlations.

### 5.3 Performance under multiple spurious correlations

Since there are many underlying attributes in the dataset, it is common that multiple spurious correlations can occur simultaneously between many attributes to a target label. A good SSL pretraining method should be robust to all these spurious correlations instead of focusing on only one of them. In this section, we evaluate the robustness of LA-SSL in the presence of multiple spurious correlations. We downsample the no-finding male patients in the training set of MIMIC-CXR by 90% to create an additional spurious correlation between gender and the symptom findings. We conduct SSL pretraining and linear probing on the synthetic training set, and evaluate the model on the original test set. Table 3 reports the test performance on inferior subgroups in both spurious attributes, male and old. LA-SSL is able to improve the accuracy in both groups. This demonstrates its capability to improve the robustness when multiple spurious correlations coexist in the dataset.

Table 4: Accuracy on the corrupted CIFAR-10 dataset of various SSL frameworks trained with and without using learning-speed aware sampling (LA-SSL).

| Method | LA-SSL | Correlation-aligned ratio ($k\%$) | | | |
|---|---|---|---|---|---|
| | | 95% | 98% | 99% | 99.5% |
| BarlowTwins | ✗ | 47.80 | 37.66 | 30.43 | 22.11 |
| | ✓ | **51.45** | **40.98** | **34.29** | **25.08** |
| SimSiam | ✗ | 21.17 | 17.72 | 15.52 | 13.49 |
| | ✓ | **24.01** | **19.38** | **16.54** | **14.73** |
| DINO | ✗ | 52.79 | 42.40 | 36.09 | 30.33 |
| | ✓ | **54.73** | **44.93** | **38.78** | **32.85** |

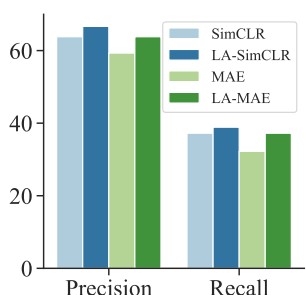
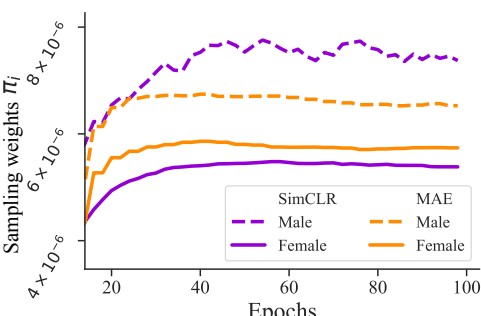

Figure 2: **Left**: Comparison of the performances on CelebA between the uniform sampling and LA-SSL versions of SimCLR and MAE. LA-SSL improves the accuracy of the male subgroup in both SSL frameworks. **Right**: The median of LA-SSL sampling probabilities for males and females with blond hair in CelebA, based on different SSL frameworks. In both frameworks, LA-SSL assigns higher probabilities to training examples in the minority group (male blonds).

## 5.4 Generalization to other SSL frameworks

In previous sections, we build the speed-aware sampling based on the similarity between two augmentations of the same image. This approach is inspired by a contrastive SSL method SimCLR. In this section, we show that the proposed method can be generalized to various SSL frameworks.

**Two views-based SSL** Despite variations in loss functions, most SSL methods with two encoder branches, such as SimCLR, BarlowTwins (Zbontar et al., 2021), SimSiam (Chen & He, 2020), and DINO (Caron et al., 2021), share a common objective to learn augmentation-invariant features maximizing the similarity between representations from two distinct views. Figure 5 in Appendix B.1 shows that The learning speed differences persist on other SSL frameworks. Hence, our proposed method can be seamlessly integrated with these SSL frameworks by monitoring similarity scores and implementing learning speed-aware sampling following Algorithm 1 during training. To assess the efficacy of the proposed approach, we conducted comparisons with alternative SSL methods on corrupted CIFAR-10 datasets, both with and without resampling. Table 4 shows that the DINO distillation method exhibits greater robustness to spurious correlations in the dataset, while the contrastive learning method without negative samples, Simsiam, is significantly affected. Despite the differences, incorporating learning speed-aware sampling consistently improves the performance under various SSL frameworks, demonstrating its general applicability.

**Generative-based SSL** The LA-SSL framework can also be adapted to generative-based SSL, such as the Masked autoencoder (MAE) (He et al., 2021), which uses a vision transformer (ViT) (Dosovitskiy et al., 2021) as the feature extractor and trains the network to reconstruct the randomly masked out patches in ViT.

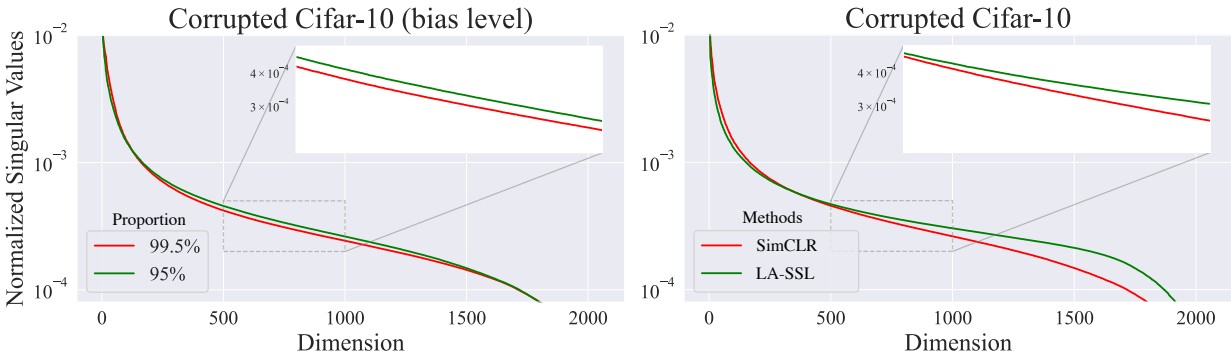

Figure 3: Spectral analysis of SSL-pretrained data representations on Corrupted CIFAR-10 using normalized singular values. The **left** plot demonstrates faster decay of singular values when the representation is trained on a dataset with stronger spurious correlation (99.5% compared to 95%). The **right** plot compares SimCLR with LA-SSL, showing slower decay of singular values in LA-SSL. See Appendix B.2 for the results on CelebA and MIMIC-CXR.

To adapt the LA-SSL framework to MAE, we monitor the learning speed based on the similarity between the representations of two randomly masked versions of the examples during training. We applied the LA-SSL equipped MAE on the CelebA dataset. The right panel of Figure 2 demonstrates that the learning speed difference persists within the MAE framework. Uniform sampling hampers training on correlational samples, which represent the minority subgroups. This observation supports the hypothesis that the learning speed difference remains a relevant factor in MAE, which can be addressed using LA-SSL. The left panel of Figure 2 shows that applying LA-SSL to MAE improves the precision and recall of the hair color classification task among males.

## 5.5 Spectral analysis on representations

To understand why LA-SSL pretrained representation can generalize better in the presence of spurious correlation, we conduct spectral analysis on the data representations of the training set. The distribution of singular values of pretrained representation is particularly important due to the *gradient starving* phenomenon when training the downstream classification tasks (Tachet et al., 2020; Pezeshki et al., 2021). This phenomenon reveals that minimizing cross-entropy loss captures only a subset of attributes relevant to the task, and the attributes captured depend on the distribution of singular values, especially when the target is correlated with certain attributes.

Suppose we train a linear classifier to classify target labels $y \in \{0, 1\}^n$ as a downstream task of SSL, we define the data representations from the feature extractor by $\Phi \in \mathbb{R}^{n \times d}$ where $\phi_i = f(x_i)$ and a linear classifier with parameters $\theta \in \mathbb{R}^d$. Then we perform a linear probing on top of this learned representation for the downstream task. Specifically, we minimize the binary cross-entropy loss $\mathcal{L}_{\text{BCE}}$ between target labels and model's prediction, defined as:

$$\mathcal{L}_{\text{BCE}} = -\sum_{i=1}^{n} y_i \log(\hat{y}_i) + (1 - y_i) \log(1 - \hat{y}_i) \tag{4}$$

where $\hat{y}_i = \sigma\left(\theta^T \phi_i\right)$ is the model prediction. To optimize the loss with gradient descent, we update the linear coefficients $\theta$ with the gradient of the $\mathcal{L}_{\text{BCE}}$ with respect to $\theta$. Denote the singular value decomposition (SVD) of $\Phi$ as $\Phi = USV^T$, the gradient becomes:

$$\frac{\partial \mathcal{L}_{\text{BCE}}}{\partial \theta} = \Phi^T\left(\hat{y} - y\right) = VSU^T\left(\hat{y} - y\right) \tag{5}$$

$S$ is a diagonal matrix, which contains the singular values that indicate the importance of singular vectors $v_i$'s. This suggests that the gradient of $\mathcal{L}_{\text{BCE}}$ is dominated by the directions of singular vectors corresponding to

higher singular values. When there is spurious correlation, the linear classifier quickly picks up the spurious attributes while unable to learn enough about task-related target labels, suggesting that those easy-to-learn attributes lie in the subspace of representations that correspond to large singular values, while the target labels are associated with subspaces corresponding to smaller singular values (Chen et al., 2019; Park et al., 2022). When training the linear classifier, the gradient becomes biased towards the subspace of spurious attributes. To reduce such bias in the gradient, a smaller gap between singular values on subspaces of representation corresponding to spurious and target attributes is desirable. We show that LA-SSL indeed results in a flatter distribution of singular values.

Figure 3 depicts the normalized singular values of data representations pretrained by SSL under various conditions. The left plot compares the singular values of data representations trained on datasets with different levels of spurious correlations. It illustrates that the top few normalized singular values of representations trained with a more biased dataset (99.5% correlation as opposed to 95%) are similar to or even greater than those of the less biased dataset. However, the remaining majority of singular values decay significantly faster in biased representations, which strongly suppresses the feature space and weakens the discriminability of other non-dominating attributes. This explains why datasets with more bias make it easier for the classifier to overfit the spurious attributes. The right plot in Figure 3 and plots in Appendix B.2 compare vanilla SimCLR with LA-SSL. The slower rate of decay in singular values supports that LA-SSL pretrained representation is more robust in classifying attributes that are impacted by the spurious correlation and other non-dominating attributes.

## 6 Conclusion

In this work, we addressed the challenge of making self-supervised learning (SSL) robust to spurious correlation. We observed that current SSL methods can be limited in scenarios with the imbalanced distribution of correlated underlying attributes, as uniform sampling may lead the model to overfit to correlation-aligned examples. This can suppress features related to attributes involved in spurious correlation. To overcome this limitation, we proposed a novel SSL framework that dynamically adjusts the sampling rates during training based on the learning speed of each example. Our method demonstrated improved performance on downstream tasks across three datasets affected by spurious correlations, showing consistent improvements across different SSL frameworks.

**Limitations** While our work highlights LA-SSL effectively mitigates spurious correlations in SSL, there are also some broader challenges for future research. LA-SSL relies on the assumption that dominant attributes are easier to learn than subdominant attributes, which does not always hold. For instance, in scenarios where two correlated attributes are both difficult to learn, their correlation may still influence the learned representations. For example, two diseases might be highly correlated but without causality. In such cases, LA-SSL may struggle to fully address the biases. This is actually a broader challenge in tackling spurious correlations for both SSL and supervised learning setting. Additionally, the lack of diverse datasets hinders better benchmarking and addressing this potential issue. In real-world datasets, the key attributes that contribute to spurious correlations may not be explicitly labeled or easily, making it difficult to identify broader spurious correlation types. These limitations highlight important directions for future work.

## Acknowledgement

This study was supported by the National Institute On Aging of the National Institutes of Health under Award R01AG079175 and R01AG085617, and NSF grant NRT-1922658.

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

## Supplementary Material

In Appendix A, we include additional descriptions of the datasets (Appendix A.1) and implementation details (Appendix A.2).

In Appendix B, we include the additional plots for learning speed analysis (Appendix B.1) and spectral analysis (Appendix B.2). In Appendix B.3, we report the model performances of all subgroups in CelebA and MIMIC-CXR. In Appendix B.4, we analyze the sensitivity of hyper-parameters in the scaling function $h$. In Appendix B.5, we compare LA-SSL with another robust SSL baseline which is developed without considering the data imbalance in spurious correlation.

## A    Experiments Settings

### A.1    Datasets

**Corrupted CIFAR-10**    (Hendrycks & Dietterich, 2019) is a synthetic dataset generated by corrupting 60,000 images in CIFAR-10 with different types of noises, where there is spurious correlation between the label and the noise type in the training and validation set, but not in the test set.

To simulate the spurious correlation in the training and validation set, a certain percentage, denoted as $k\%$, images are corrupted with a specific noise type that matches their label, while the other $1 - k\%$ of images in each class are randomly corrupted with different noise types. The higher the value of $k$ is, the stronger the correlation between noise types and labels. The following are the numbers of images corrupted with label-related noise types versus random noise types for different values of $k$: 44,832 vs 228 for $k = 99.5$, 44,527 vs 442 for $k = 99$, 44,145 vs 887 for $k = 98$, and 42,820 vs 2,242 for $k = 95$. All the images in the test set are randomly corrupted with different noise types.

In this study, we adopt the simulation settings from a previous paper (Nam et al., 2020). We include the following corruption types for Corrupted CIFAR-10 dataset: *Brightness*, *Contrast*, *Gaussian Noise*, *Frost*, *Elastic Transform*, *Gaussian Blur*, *Defocus Blur*, *Impulse Noise*, *Saturate*, and *Pixelate*. Each corruption has 5 strength level settings in the original paper (Hendrycks & Dietterich, 2019) among which we apply the strongest level. In the simulation, these types of corruptions are highly correlated with the original classes of CIFAR-10, which are *Plane*, *Car*, *Bird*, *Cat*, *Deer*, *Dog*, *Frog*, *Horse*, *Ship*, and *Truck*.

**CelebA**    (Liu et al., 2015) is a dataset that contains 202,599 images of celebrities along with various attributes associated with the images. One attribute indicates whether the person has *Blond Hair*, which exhibits a strong spurious correlation with the *Male* attribute. In the dataset, 38.64% of the images are males, and 61.36% are females. However, only 2% of males in the dataset has blond hair as opposed to 24% in females. The hair color classification is the downstream task we evaluate. We use the original training, validation and test set split in the dataset, which results in 162,770, 19,867, and 19,962 samples, respectively.

**MIMIC-CXR**    (Johnson et al., 2019) is a dataset of 377,110 chest X-ray images labeled with the diagnosis. For our analysis, we focused on images with demographic information and in anteroposterior (AP) and posteroanterior (PA) positions, resulting in 228,905 X-ray images. We split these samples into the training, validation, and test set by patients at an 8:1:1 ratio.

Our downstream task is to classify whether the patients exhibit any medical findings based on their chest X-ray images. The *No Findings* attribute and the *Age* attribute are correlated. Among younger patients, the *No Findings* attribute is more prevalent, while older patients are more likely to have some form of disease or abnormality. We divided patients into two sub-cohorts based on whether they were younger or older than 90. Among the younger group, 32.43% of patients have no findings, while only 16.07% of older patients exhibit no findings.

## A.2 Implementation details

All experiments were conducted on NVIDIA RTX8000 GPUs and NVIDIA V100 GPUs. The settings of the model training on different datasets are listed below:

**Corrupted CIFAR-10**

We use SimCLR (Chen et al., 2020) to train a Resnet50 feature extractor for 2000 epochs, at which we notice that the training loss converges. We set the batch size to 1024, the projection head size to 512 and the temperature to 0.5. We use SGD with a learning rate of 0.5, weight decay of $10^{-4}$, 50 warmup epochs, and a cosine annealing scheduler. We apply random data augmentation to each image, including:

- Random crop with scale $[0.2, 1.0]$,

- Random horizontal/vertical flipping with 0.5 probability,

- Random ($p = 0.8$) color jittering: brightness, contrast, and saturation factors are uniformly sampled from $[0.8, 1.2]$, hue factor is uniformly sampled from $[-0.1, 0.1]$,

- Random grayscale ($p = 0.2$),

- Random solarization ($p = 0.2$).

For the LA-SSL method, we adopted the same settings as SimCLR. Additionally, we set the scale $\gamma = 10$, threshold $r = 0.01$ and update $\pi$ every 20 epochs.

**CelebA** We trained SimCLR using the same hyperparameters as Corrupted CIFAR-10. In the training process, we use a batch size of 512, warmup epochs at 10 and train the SSL model for 200 epochs. We apply random data augmentation to each image, including:

- random crop with scale $[0.2, 1.0]$,

- random horizontal flipping with 0.5 probability,

- random ($p = 0.8$) color jittering: brightness, contrast, and saturation factors are uniformly sampled from $[0.6, 1.4]$, hue factor is uniformly sampled from $[-0.1, 0.1]$,

- random Gaussian blur ($p = 0.5$),

- random grayscale ($p = 0.2$),

- random solarization ($p = 0.2$).

For the LA-SSL method, we adopted the same settings as SimCLR. Additionally, we set the scale $\gamma = 10$, threshold $r = 0.1$ and update $\pi$ every 2 epochs.

**MIMIC-CXR** We trained SimCLR using the same hyperparameters as Corrupted CIFAR-10. In the training process, we use a batch size of 512, warmup epochs at 10 and train the SSL model for 100 epochs. We apply random data augmentation to each image, including:

- random rotation with a degree uniformly sampled from $[0, 30]$

- random crop with scale $[0.7, 1.0]$ to size at $224 \times 224$,

- random horizontal flipping with 0.5 probability,

- random ($p = 0.8$) color jittering: brightness, contrast, and saturation factors are uniformly sampled from $[0.6, 1.4]$, hue factor is uniformly sampled from $[-0.1, 0.1]$,

- random grayscale ($p = 0.2$),

- random Gaussian blur ($p = 0.5$),

- random solarization ($p = 0.2$).

For the LA-SSL method, we adopted the same settings as SimCLR. Additionally, we set the scale $\gamma = 10$, threshold $r = 0.1$ and update $\pi$ every 2 epochs.

## B  Additional Results

### B.1  Learning speed

In Figure 4, we show the training dynamics of SimCLR and LA-SSL on corrupted CIFAR-10 at varying levels of spurious correlation ($k\%$) in supplementary to Figure 1. The plot shows that the feature extractor in SimCLR consistently learns faster on the correlation-aligned examples than on the correlation-conflicting examples under different levels of spurious correlation. The reweighted sampling approach in LA-SSL consistently narrows the gap between conflict and aligned examples by upsampling the examples that learn slower. Figure 5 shows that The learning speed differences persist on other SSL frameworks. Leveraging the reweighted sampling mitigates the gap on other SSL methods as well.

### B.2  Spectral analysis

In Figure 6, we plot the normalized singular values of data representations pretrained by SSL on all three datasets in supplement to Figure 3. They compare vanilla SimCLR with LA-SSL and demonstrate that the rate of decay in singular values of LA-SSL is slower consistently on all three datasets.

### B.3  Results on subgroups

Table 5 and 6 report the performances of each subgroup in CelebA and MIMIC-CXR. LA-SSL is able to improve the model performance on the group with inferior performance while maintaining the performance on the superior subgroups.

Table 5: Downstream task performance for all the subgroups regarding gender in the CelebA dataset on hair color classification.

| Method | Prevalance | | AUC | | Precision | | Recall | |
|---|---|---|---|---|---|---|---|---|
| | Female | Male | Female | Male | Female | Male | Female | Male |
| SimCLR | 20.24 | 2.33 | 97.91 | 95.14 | 81.86 | 63.80 | 88.66 | 37.22 |
| LA-SSL | 20.24 | 2.33 | 97.89 | **95.63** | 81.54 | **66.66** | 90.00 | **38.88** |

Table 6: Downstream task performance for all the subgroups regarding age in MIMIC-CXR dataset on No Finding classification

| Method | Prevalance | | AUC | | Precision | | Precision | |
|---|---|---|---|---|---|---|---|---|
| | Young | Old | Young | Old | Young | Old | Young | Old |
| SimCLR | 32.43 | 16.07 | 82.25 | 73.75 | 62.33 | 37.40 | 72.25 | 31.61 |
| LA-SSL | 32.43 | 16.07 | 83.47 | 74.89 | 63.65 | **40.00** | 73.42 | **34.83** |

### B.4  Sensitivity analysis on the scaling function $h$

In Section 4.3, we define a scaling function to enlarge the gap of learning speeds among different samples, which include two hyperparameters: scale $\gamma$ and threshold $r$. We conduct the analysis on how sensitive

Table 7: Comparison among SimCLR, LA-SSL, and contrastive learning with adversarial perturbation on corrupted CIFAR-10 (95%). LA-SSL is able to obtain significantly more improvements compared to adversarial perturbation technique.

| Method | SimCLR | Adversarial Perturbation ($\epsilon$) | | | LA-SSL |
| --- | --- | --- | --- | --- | --- |
| | | 0.05 | 0.1 | 0.2 | |
| Accuracy | 44.08 | 44.72 | 45.41 | 45.26 | **48.02** |

the model to $h$ on CelebA. We experiment with various values of $r$ while keeping $\gamma$ fixed at 10. Similarly, we investigate the effect of different $\gamma$ values when $r$ is set to 0.1. Figure 7 illustrates that the model's performance remains relatively consistent across different $r$ values within a reasonable range. However, when $\gamma$ is set to 5, the scale becomes too small and the performance deteriorates.

### B.5 Comparison with other robust SSL baselines

Some other SSL methods have demonstrated the ability to mitigate the issue of learning shortcuts during training (Robinson et al., 2021; Lin et al., 2022; Hamidieh et al., 2022), which could potentially enhance robustness under spurious correlations. In our experiments, we explored one such method that has a publicly available implementation. Robinson et al. proposed a technique that incorporates adversarial perturbations into the contrastive loss to overcome feature suppression (Robinson et al., 2021). Table 7 shows the adversarial perturbation does improve the SimCLR baseline. However, compared to LA-SSL, the improvements achieved by this method are relatively limited. Similar findings have also been reported in (Park et al., 2022). This could be attributed to the fact that robust SSL methods designed for general purposes may not specifically address the attribute imbalance among different subgroups.

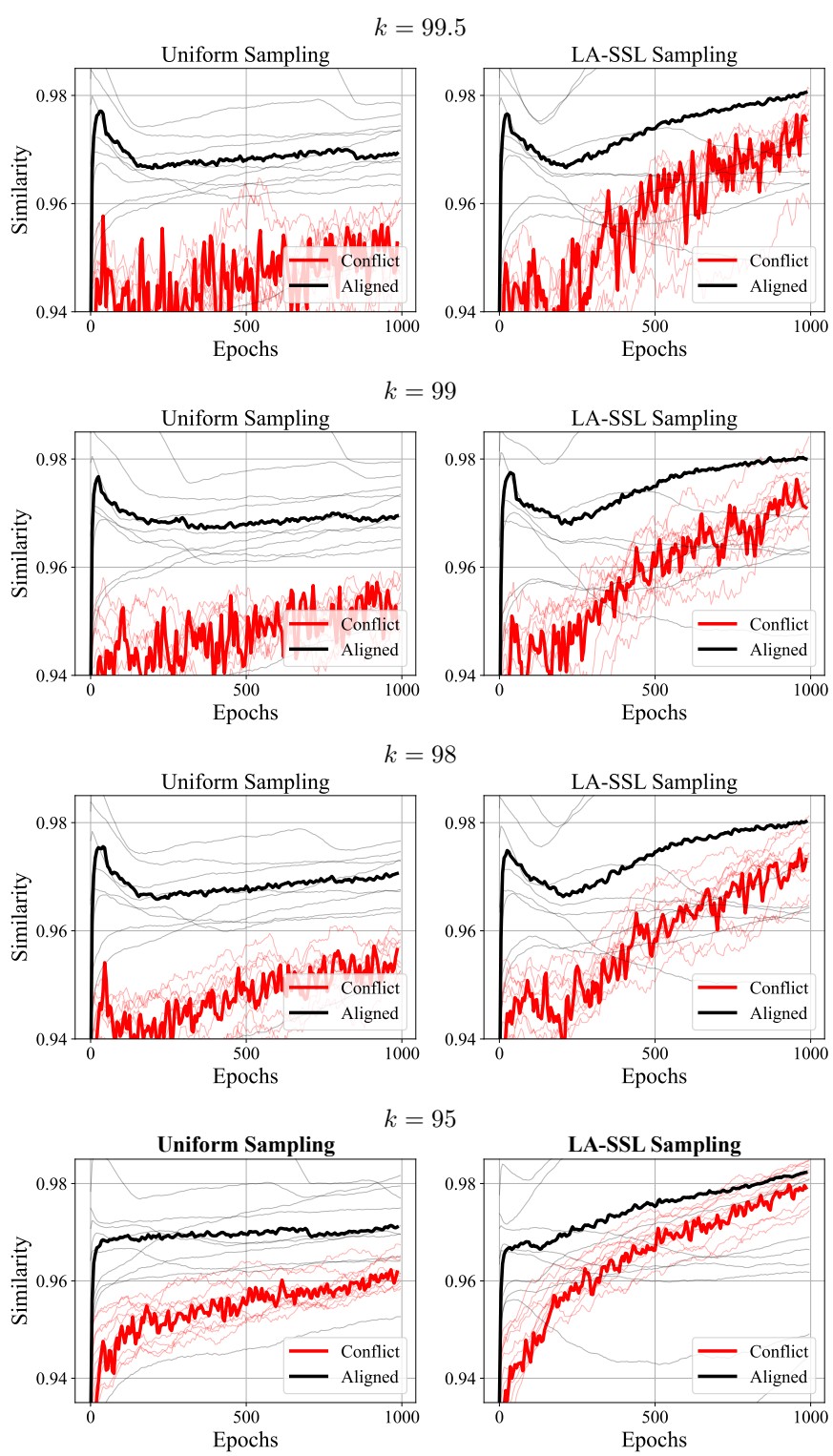

Figure 4: The similarity between the representations of two augmented views on corrupted CIFAR-10 at varying level of spurious correlations during training. The thick curves represent the mean of similarities of correlation-conflicting and -aligned examples, while the light curves represent the similarity mean of each class.

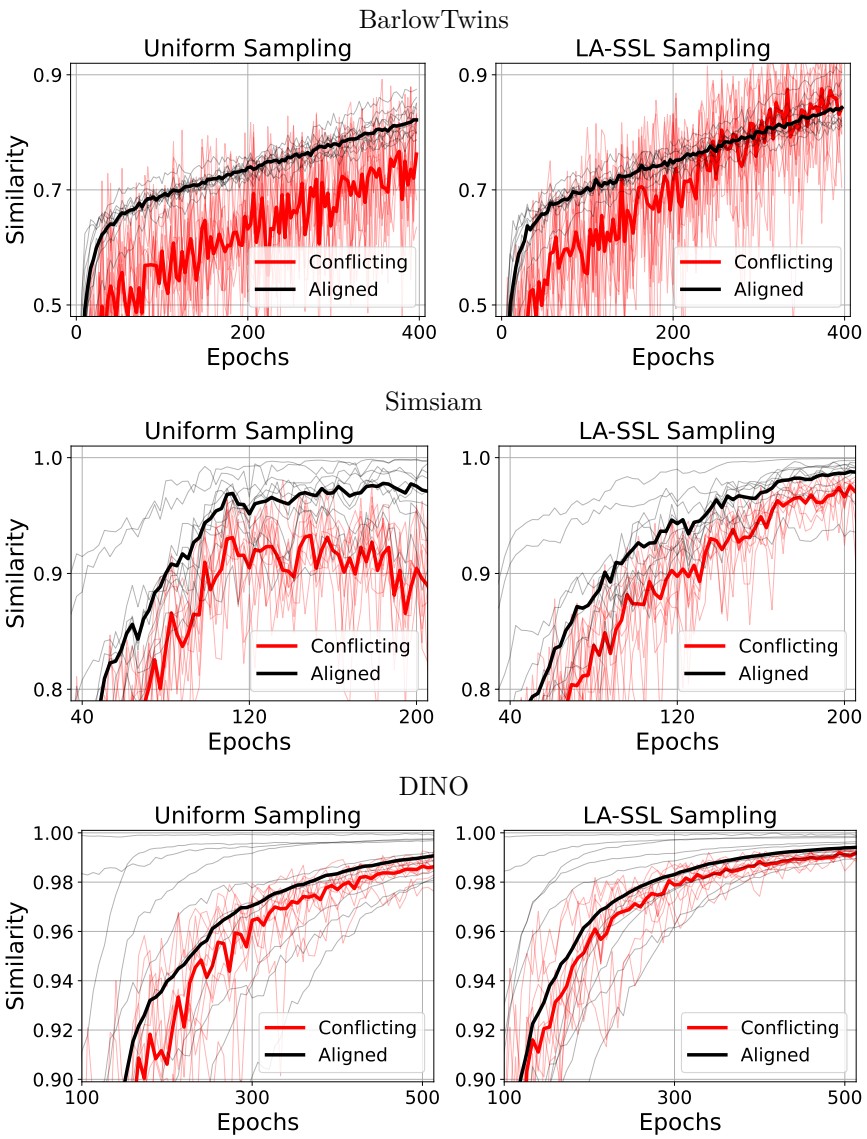

Figure 5: The similarity between the representations of two augmented views on corrupted CIFAR-10 under different SSL frameworks. The thick curves represent the mean of similarities of correlation-conflicting and -aligned examples, while the light curves represent the similarity mean of each class.

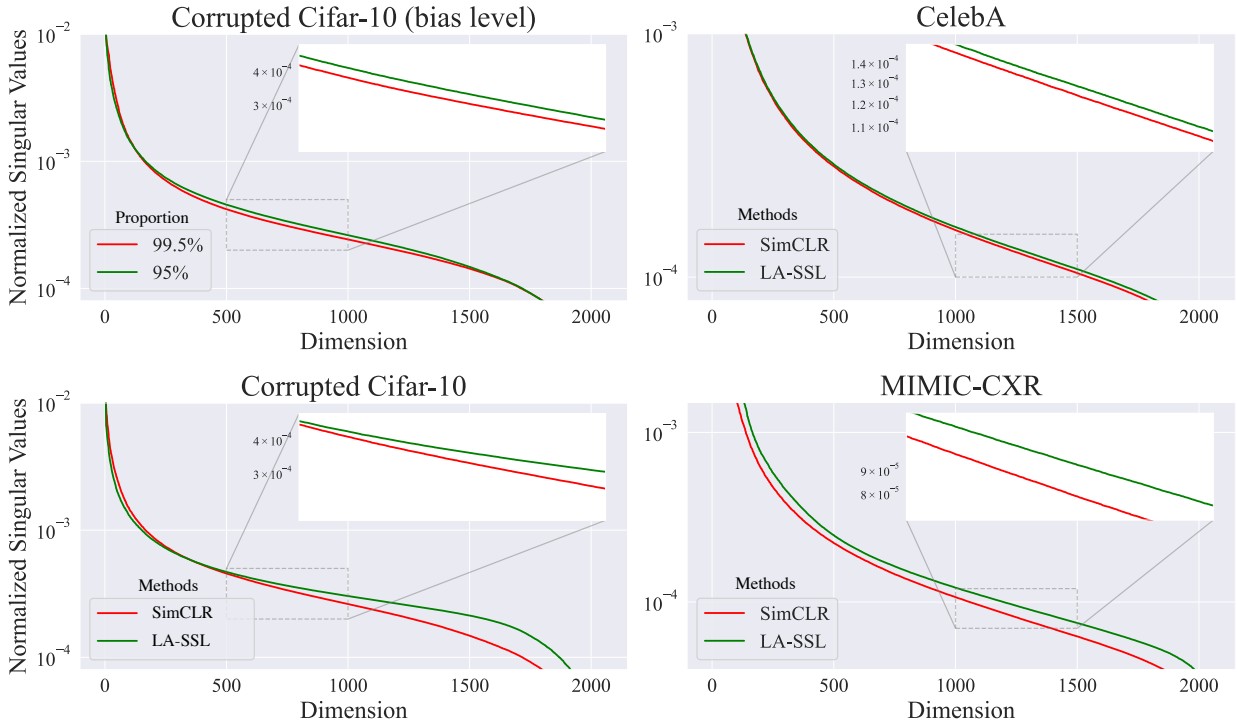

Figure 6: The **left** column shows the normalized singular values of representation pretrained on Corrputed CIFAR-10. The **top-left** plot shows when the representation is trained on the dataset stronger spurious correlation (99.5% opposed to 95%) between the singular values decays faster. The **top-left** plot compares SimCLR with LA-SSL. The singular values of LA-SSL decay slower. The **right** column indicates similar trends on two real-world datasets. LA-SSL enables singular values to decay slower.

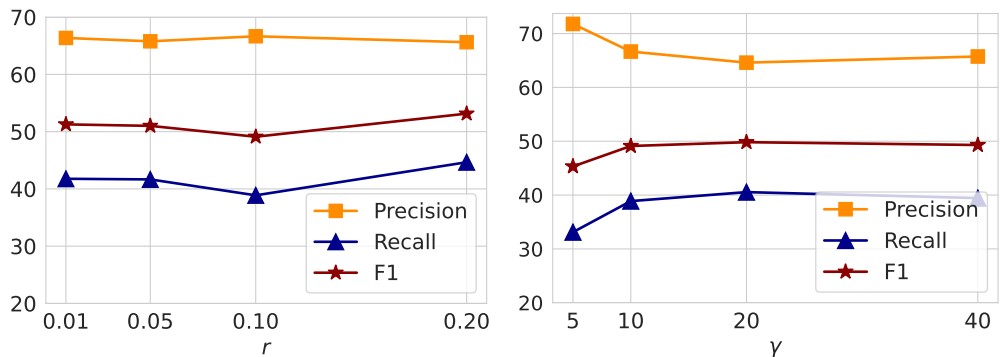

Figure 7: The sensitivity analysis on $r$ and $\gamma$ in function $h$ on CelebA.

