# OpenReview forum: "Making Self-supervised Learning Robust to Spurious Correlation via Learning-speed Aware Sampling"
_TMLR — Accepted by TMLR_

### Review · Reviewer_7D2K · 2024-10-15

**Summary Of Contributions:**

The authors study the problem of improving the robustness of self-supervised learning (SSL) algorithms and dealing with spurious correlations in the training data. In particular, core contributions from the paper include:
1. **instance-level learning dynamics**: analysis showing that current SSL methods can overfit spurious correlations between underlying attributes in imbalanced datasets. The SSL loss can be minimized by only learning features related to some easy-to-learn attributes while suppressing 'hard to learn' and discriminative features. The authors also observe that samples whose attributes do not align with spurious correlations ("conflicting" samples) are learned slower by SSL models compared to "aligned" samples. This learning speed difference is leveraged as a proxy for the spurious attributes.

2. **learning-aware SSL**: This is a proposal for a learning-speed-aware sampling method for SSL (LA-SSL). It dynamically adjusts the sampling probability for each example inversely to its learning speed. This upsamples conflicting examples to force learning more diverse features.

3. **benchmarks**: demonstration that LA-SSL improves the robustness of learned representations on three image datasets with spurious correlations:
   - Corrupted CIFAR-10 (synthetic spurious correlation)
   - CelebA (spurious correlation between hair color and gender)
   - MIMIC-CXR (spurious correlation between disease findings and age)

4. Analysis showing LA-SSL is robust to multiple simultaneous spurious correlations and generalizes across SSL frameworks (SimCLR, BarlowTwins, SimSiam, DINO, MAE). Spectral analysis indicates that LA-SSL leads to a slower decay of singular values of the learned representation compared to standard SSL, enabling better discriminative power for attributes impacted by spurious correlation.

**Audience:**

Yes

**Broader Impact Concerns:**

The paper proposes LA-SSL, a technical approach to mitigate the impact of spurious correlations on self-supervised learning (SSL) representations, aiming to improve their fairness and robustness across different subgroups in the data. No significant ethical concerns requiring a separate Broader Impact Statement were identified, but the paper could benefit from a brief discussion on the responsible use and limitations of LA-SSL to guide practitioners and researchers.

**Claims And Evidence:**

Yes

**Requested Changes:**

**Adjustments Critical for Acceptance**
* Clarify the novelty and significance of the contribution in the context of existing work on mitigating spurious correlations and improving SSL robustness. While the paper demonstrates the effectiveness of LA-SSL, it would be essential to discuss how it advances the state-of-the-art and differs from related techniques.
* Include experiments that use a more complex dataset (in terms of the number of classes/scale), like the ImageNet-100 dataset, or more complex spurious correlations (in terms of aligned vs conflicted subgroups) to demonstrate the scalability and effectiveness of LA-SSL. This is critical to show that the proposed method can handle the challenges of larger-scale datasets and more diverse spurious correlations.


**Adjustments to Strengthen the Work**
* Include comparisons to additional baselines, such as other techniques proposed for mitigating spurious correlations or improving SSL robustness. This would help better situate the contribution and demonstrate the advantages of LA-SSL over existing approaches.
* Discuss the computational overhead introduced by LA-SSL compared to standard SSL training. While dynamic sampling is a key aspect of the approach, it would be informative to understand its impact on training time and resources.
* Provide a more comprehensive discussion on the limitations and potential failure cases of LA-SSL. While the experiments demonstrate its effectiveness, understanding the scenarios where it may not perform well would help characterize its applicability and guide future work.

**Strengths And Weaknesses:**

**Strengths**
+ Novel and generalizable approach across SSL algorithms: The paper proposes a novel learning-speed aware sampling method (LA-SSL) to mitigate the impact of spurious correlations on learned representations in self-supervised learning (SSL). The approach is shown to generalize across multiple SSL frameworks like SimCLR/BarlowTwins/DiNO.
+ Insightful analysis: The authors provide an insightful analysis of how SSL models can overfit spurious correlations by learning features related to easy-to-learn attributes while suppressing others. They also show that samples defying the spurious correlations are learned more slowly, which is leveraged in the LA-SSL algorithm.
+ Spectral analysis: The paper includes a spectral analysis of the learned representations, providing a theoretical justification for why LA-SSL enables better discriminative power for attributes impacted by spurious correlations.


**Potential Areas for Improvement**
+ Comparison to more baselines: While the paper compares LA-SSL to standard SSL approaches, it would be informative to include comparisons to other techniques proposed for mitigating spurious correlations or improving SSL robustness to strengthen the core contributions of the paper.
+ Computational cost: The paper could discuss the computational overhead introduced by LA-SSL compared to standard SSL training, as the dynamic sampling may impact training time. This is particularly important in the context of large-data training.
+ Limitations and failure cases: Discussing any observed limitations or failure cases of LA-SSL would help provide a more complete picture of the technique's applicability and areas for future improvement.
+ Scaling to more classes/datasets: The authors already include several benchmarks like CIFAR/Mimic. Including some analysis on a larger number of classes (like Imagenet-100) and more complex spurious relations would help.

---

> ### Author Response · Authors · 2024-11-21
>
> Thank you for your review. Here are our responses.
>
> **C1. Comparison to more baselines: While the paper compares LA-SSL to standard SSL approaches, it would be informative to include comparisons to other techniques proposed for mitigating spurious correlations or improving SSL robustness to strengthen the core contributions of the paper.**
>
> **1. Clarify the novelty and significance of the contribution in the context of existing work on mitigating spurious correlations and improving SSL robustness. While the paper demonstrates the effectiveness of LA-SSL, it would be essential to discuss how it advances the state-of-the-art and differs from related techniques.**
>
> Thanks for your suggestion. We discussed both works on mitigating spurious correlation and improving SSL robustness in Section 2 and delineated our contribution. Compared to supervised methods in mitigating spurious correlation, we emphasize that our approach “focuses on learning robust representations from unlabeled datasets, which is distinct from the de-biasing algorithms in the supervised learning setting“. Compared to robustness SSL method, we focus on “analyzing and improving the robustness of SSL pretraining for the downstream tasks where spurious correlation persists” and “specifically address the data imbalance caused by spurious correlation”. These distinctions clarify the unique problem setting and motivation of our work compared to prior studies.
>
> Additionally, in Section 5.2, we revised the paper to include a direct comparison of LA-SSL with two baselines: one supervised method designed to mitigate spurious correlations and one robust SSL method. The results demonstrate that LA-SSL achieves superior performance in these scenarios, further underscoring its effectiveness.
>
>
> **2. Include comparisons to additional baselines, such as other techniques proposed for mitigating spurious correlations or improving SSL robustness. This would help better situate the contribution and demonstrate the advantages of LA-SSL over existing approaches.**
>
> This is a good point. In Table 4, we compared the performance across different SSL method such as BarlowTwins, SimSiam and DINO. DINO and BarlowTwins themselves are more robust SSL methods than SimCLR and LA-SSL can further improve upon them. In addition, we compared with a method to mitigate shortcut learning [1] in Appendix B.5 developed based on SimCLR,  showing that LA-SSL achieves a better performance. We added a discussion of these results in Section 5.2. We also compared LA-SSL with some supervised debiasing methods in Section 5.2, demonstrating that LA-SSL “achieves even better performance than existing supervised de-biasing algorithms for spurious correlations”.
>
>
> **C2. Computational cost: The paper could discuss the computational overhead introduced by LA-SSL compared to standard SSL training, as the dynamic sampling may impact training time. This is particularly important in the context of large-data training.**
>
> Thanks for pointing this out. Computational efficiency is actually a strength of LA-SSL. Since the similarity scores are computed directly from the representation of two augmented views (Algorithm 1 Line 5) and updated on-fly (Line 7), it only requires O(d) extra computation for each sample where d is the dimension of the representations. It is negligible compared to model training. Also, the adjustment in sampling requires no extra computation.
>
> **C3. Limitations and failure cases: Discussing any observed limitations or failure cases of LA-SSL would help provide a more complete picture of the technique's applicability and areas for future improvement.**
>
> Thank you for the valuable suggestion. We agree that discussing limitations is crucial to provide a comprehensive view of LA-SSL and to inspire future research. To address this, we have added a Limitations section, highlighting two primary challenges:
>
> -  Identification of Spurious Correlations: While datasets with spurious correlations are prevalent across various domains, identifying them can be challenging due to limited auxiliary attribute information. This constraint poses a barrier to further research on SSL methods addressing spurious correlations, as the necessary data for validation and benchmarking is often unavailable.
>
> - Dependency on Learning-Speed Differences: LA-SSL leverages learning-speed differences between dominant and subdominant attributes. This approach is effective when the dominant attributes are significantly easier to learn. However, in cases where two correlated attributes are both difficult to learn, their interactions may still influence the model, but LA-SSL may not adequately address such scenarios. This limitation highlights the broader challenge in research on spurious correlations, which is further hindered by the lack of suitable benchmark datasets.
>
> We acknowledge these limitations as areas where further exploration is needed and hope they will motivate future studies to address these gaps.

---

> > ### Author Response · Authors · 2024-11-21
> >
> > **C4. Include experiments that use a more complex dataset (in terms of the number of classes/scale), like the ImageNet-100 dataset, or more complex spurious correlations (in terms of aligned vs conflicted subgroups) to demonstrate the scalability and effectiveness of LA-SSL. This is critical to show that the proposed method can handle the challenges of larger-scale datasets and more diverse spurious correlations.**
> >
> > We agree that testing SSL algorithms on a more complex dataset, e.g. ImageNet, would be very desirable. However, the attributes of images are not recorded. Consequently, it is hard to define and find explicit spurious correlations. Previous studies on spurious correlation in supervised learning are based on even smaller datasets, such as C-MNIST and Waterbird (which only contains simple features with thousands of images), partially due to the limitation of defining spurious correlation on large natural image datasets. Therefore, we can only focus on these benchmark datasets with rich attribute information for investigating spurious correlations. In terms of scale of the dataset, MIMIC-CXR and CelebA actually consist of a larger number of samples than ImageNet-100.
> >
> > Given the limitation of the data, we also attempted to analyze the performance on more complex spurious relationships. In Table 3, we presented a synthetic version of MIMIC-CXR, where the spurious correlation between age, gender and disease co-occurs. Such “two-level” spurious correlation was not even well studied in supervised learning literature. The superior performance of LA-SSL demonstrates that it can also mitigate more complex spurious correlations.
> >
> > **C5. Discuss the computational overhead introduced by LA-SSL compared to standard SSL training. While dynamic sampling is a key aspect of the approach, it would be informative to understand its impact on training time and resources.**
> >
> > Thanks for the advice. To address C2 and C5, we added a discussion on the computation overhead in Section 4.3.
> >
> > **C6. Provide a more comprehensive discussion on the limitations and potential failure cases of LA-SSL. While the experiments demonstrate its effectiveness, understanding the scenarios where it may not perform well would help characterize its applicability and guide future work.**
> >
> > To address C3 and C6, we added a Limitation section in Section 6 to present the limitations of the current study and some potential directions for further study.
> >
> > [1] Joshua Robinson, Li Sun, Ke Yu, K. Batmanghelich, Stefanie Jegelka, and Suvrit Sra. Can contrastive learning avoid shortcut solutions? Advances in neural information processing systems, 34:4974–4986, 2021

---

### Review · Reviewer_DRo7 · 2024-11-05

**Summary Of Contributions:**

This paper introduces a novel method to address the presence of spurious correlations in representation learning for self-supervised learning objectives like SimCLR. To do so, the authors use a "learning-speed aware" method to upsample data points that are harder to learn, as measured by their similarity scores (when encoded after applying two random augmentations). The method is evaluated on three datasets—Corrupted CIFAR-10, CelebA, and MIMIC-CXR—showing improved robustness and performance in downstream tasks. The approach is also extended to various SSL frameworks beyond SimCLR, and spectral analysis is provided to explain the effectiveness of the method.

**Audience:**

Yes

**Broader Impact Concerns:**

No broader impact concerns.

**Claims And Evidence:**

Yes

**Requested Changes:**

1. The authors could be more explicit about the role of the singular directions in the spectral analysis, which includes picking up important features for classification (this one is critical ). Overall, I feel that this analysis is very superficial and provides little to no insight on why the method works well. A more precise analysis that might try to distinguish between the two cases (spurious correlations v.s. not) would significantly enhance the paper (this one is a suggestion).
2. An ablation study on the hyperparameters of the method, specifically the impact of $\gamma$ and $r$ in the classification accuracy could provide further insights about the robustness of the method (this one is a suggestion).

**Strengths And Weaknesses:**

Overall I believe all the claims the authors make in their paper are correct and justified with the exception of the theoretical analysis, where there are very minor gaps. I also think that based on the subject of the paper, there will be an audience interested in mitigating spurious correlation for self-supervised methods.  Based on these two criteria I believe this paper should be accepted.

**Strengths:**
1. Well written, easy to understand and well presented paper.
2. Simple method, with intuitive motivation. The method due to its simplicity is easy to implement and has negligible computational overhead.
3. Strong results, where the proposed method consistently improves the metrics across a variety of datasets and using different SSL frameworks.

**Weaknesses:**
1. Memory overhead: The method requires storing the similarity score for each datapoint in the dataset (which is not terribly bad).
2. In the spectral analysis, while it is true that singular directions will pick the easy to learn attributes, it is not clear to me that this corresponds uniquely to spurious correlations. In other words, the higher singular values will also pick important features for classification.

---

> ### Author Response · Authors · 2024-11-21
>
> Thank you for your review. Here are our responses to your comments.
>
> **C1. Memory overhead: The method requires storing the similarity score for each datapoint in the dataset (which is not terribly bad).**
>
> This is a good point. The LA-SSL algorithm requires storing the similarity score of each training sample which takes O(n) space. We add more explanations on memory and computational cost in Section 4.3. Memory overhead can be optimized by saving on CPU memory (even disk) instead of GPU memory.
>
> **C2. In the spectral analysis, while it is true that singular directions will pick the easy to learn attributes, it is not clear to me that this corresponds uniquely to spurious correlations. In other words, the higher singular values will also pick important features for classification.**
>
> We completely agree that higher singular values can capture important features. For example, in the CelebA dataset, gender is a relatively easy-to-learn feature that higher singular values often encode. This can be valuable when the task involves classifying gender-related labels. However, LA-SSL actually does not make the representation invariant to, or less discriminative for these important attributes. Instead, LA-SSL turns out to flatten the singular value curves and improve the discriminability of learned representation on the attributes associated with the eigenvectors of lower singular values [1]. This helps downstream classifiers capture features that might otherwise be overlooked by vanilla SSL methods.
>
> **C3.The authors could be more explicit about the role of the singular directions in the spectral analysis, which includes picking up important features for classification (this one is critical ). Overall, I feel that this analysis is very superficial and provides little to no insight on why the method works well. A more precise analysis that might try to distinguish between the two cases (spurious correlations v.s. not) would significantly enhance the paper (this one is a suggestion).**
>
> Thanks for the great suggestion. We revised Section 5.5 and added a paragraph when explaining the problem in Section 3. We hope this explains how singular values impact the discriminability on different attributes more explicitly.
>
> We also like the idea of comparing the singular values between the two cases where the dataset is with and without spurious correlations. We did analyze the singular values when the representations are trained with datasets with different strengths of spurious correlations in the left panel of Figure 3. It demonstrates that under stronger spurious correlation, a sharper distribution of singular values strongly suppresses the discriminability of the representation with respect to non-dominating attributes.
>
> **C4. An ablation study on the hyperparameters of the method, specifically the impact of γ and r  in the classification accuracy could provide further insights about the robustness of the method (this one is a suggestion).**
>
> We agree that the sensitivity of hyperparameters is crucial to the generalizability of a method. To address this, we conducted a sensitivity analysis on $\gamma$ and $r$, as presented in Appendix B.4, Figure 7, to illustrate how performance varies with these parameters. The results show that the model’s performance remains relatively stable within a reasonable range of values. However, we observed that when $\gamma$ is set to 5, the scale becomes too small, leading to a noticeable decline in performance.
>
>
> [1] Xinyang Chen, Sinan Wang, Mingsheng Long, and Jianmin Wang. Transferability vs. discriminability: Batch spectral penalization for adversarial domain adaptation. Proceedings of the 36th International Conference on Machine Learning.

---

### Review · Reviewer_BNof · 2024-11-11

**Summary Of Contributions:**

This paper studies self-supervised learning (SSL) when there are spurious correlations present. They demonstrate that SSL loss can often be minimized by capturing the subset of conspicuous features that are correlated with sensitive attributes and not identifying the important/discriminative features for downstream tasks. To combat this issue, the paper uses a learning-speed aware approach (LA-SSL) which samples data points with probability inversely proportional to its learning speed. They evaluate their method on 3 datasets and showcase the utility of their method.

**Audience:**

Yes

**Broader Impact Concerns:**

no concerns.

**Claims And Evidence:**

Yes

**Requested Changes:**

- Do the authors think that section 5.5 could be moved to earlier in the work? I find it odd that all the experimental results are presented and then the paper jumps back to some theory.

**Strengths And Weaknesses:**

Strengths:
- The authors do a nice job at clearly motivating the problem
- The related work section is well written and extensive.
- There are many different experiments that all highlight how LA-SSL outperforms the other method SimCLR.
- I really enjoy section 5.5 because it provides some intuition above why LA-SSL can generalize better.

Weaknesses:
- I am not sure I entirely understand the preliminary setting. There are underlying attributes Z_1,..., Z_p, but what does it mean they can be discretized into K_1, .., K_p categories. Is it saying that each Z_i can be discretized into K_i respective categories. Thus each attribute can be discretized but the number of categories can be different for each attribute? Is that correct?
- Could you elaborate on what this sentence means, "In such situations, the learned representations may succeed in classifying labels relying on Zi in downstream tasks, but they may fail to classify labels relying on Zj accurately.
- I think the content of section 5.5 is very compelling, but the placement of it is a little odd. I think it would make more sense earlier in the work?
- My main concern is that all of these results are empirical. Is there any theory that guarantees this idea of using learning speed is the right one? Are their scenarios where doing this actually results in worse results? If so, could the authors elaborate on this.

---

> ### Author Response · Authors · 2024-11-21
>
> Thank you for your review. Here are our responses to your questions.
>
> **C1. Is it saying that each Z_i can be discretized into K_i respective categories. Thus each attribute can be discretized but the number of categories can be different for each attribute? Is that correct?**
>
> Yes, it says each attribute is modeled as a discrete random variable with various numbers of classes. We clarified this in the updated version.
>
>
> **C2. Could you elaborate on what this sentence means, "In such situations, the learned representations may succeed in classifying labels relying on Zi in downstream tasks, but they may fail to classify labels relying on Zj accurately?**
>
> Thanks for bringing up the confusion. We revised and restructured the corresponding paragraph in Section 3 to clarify and further elaborate this sentence.
>
> Here is a more detailed explanation with an example. In the context of predicting hair color in the CelebA dataset, we can consider the attribute $Z_i$ to represent “gender” and the attribute $Z_j$ to represent “hair color”. In downstream tasks, successfully classifying “gender” requires the model to effectively capture features related to $Z_i$, while classifying “hair color” relies on the capability of the model to distinguish $Z_j$. However, the SSL model tends to learn features that can better discriminate “gender” but not “hair color”. As a result, the learned representation can perform well in downstream tasks that rely on $Z_i$ (gender) but struggle to accurately classify labels dependent on attribute $Z_j$ (hair color).
>
> **C3. I think the content of section 5.5 is very compelling, but the placement of it is a little odd. I think it would make more sense earlier in the work?**
>
> Thank you for your valuable suggestion. Section 5.5 provides a quantitative analysis of the pretrained representations from SSL methods to explain why LA-SSL achieves superior performance, as demonstrated in Section 5.2. We agree that incorporating some of this content earlier in the paper would help provide theoretical intuitions and improve the flow. Therefore, in Section 3, we added a high-level overview of spectral analysis. This addition highlights the connection between singular values and the features learned by SSL methods while referencing Section 5.5 for further details. We hope this revision enhances the coherence and readability of the paper.
>
> **C4. My main concern is that all of these results are empirical. Is there any theory that guarantees this idea of using learning speed is the right one? Are their scenarios where doing this actually results in worse results? If so, could the authors elaborate on this.**
>
> This is a good point. While our work is primarily empirical, it builds on the intuition that learning speed reflects the model’s prioritization of simpler or more dominant patterns during training. This aligns with previous studies on mitigating spurious correlation in supervised setting [1]. These concepts suggest that attributes learned faster often dominate the representation space, which can hinder the discovery of less salient yet important features.
>
> We also agree on the importance of discussing limitations. We discussed some scenarios where LA-SSL may fail - LA-SSL leverages learning speed differences between dominant and subdominant attributes. This approach is effective when the dominant attributes are significantly easier to learn. However, in cases where two correlated attributes are both difficult to learn, their interactions may still influence the model, but LA-SSL may not adequately address such scenarios. This limitation highlights the broader challenges in research on spurious correlations, which is further hindered by the lack of suitable benchmark datasets.
>
> [1] Jun Hyun Nam, Hyuntak Cha, Sungsoo Ahn, Jaeho Lee, and Jinwoo Shin. Learning from failure: De-biasing classifier from biased classifier. In Neural Information Processing Systems, 2020.

---

### Author Response · Authors · 2024-11-21

We would like to thank the reviewers for their thoughtful comments and the editor for handling our submission.

We are encouraged by your recognition of our contribution in addressing the well-motivated challenge of spurious correlations in SSL. We appreciate the acknowledgment of our novel algorithm, its validation across multiple datasets, and our spectral analysis that provides insights into better generalization.

Here, we summarize our revisions to address common questions before addressing individual comments:

**Positioning and Explanation of Spectral Analysis**

We have revised Section 5.5 for better clarity and introduced the motivation and function of the spectral analysis earlier in Section 3, where we position the problem of spurious correlation in SSL.

**Discussion of Limitations**

We recognize the importance of discussing limitations more thoroughly. To address this, we have added a dedicated section discussing potential scenarios where LA-SSL may not perform as expected, as well as the challenge posed by the limited availability of datasets to explore a wider range of spurious correlation types.

**Comparison with Additional Baselines and Datasets**

We have expanded our comparison in Section 5.2 to include other methods aimed at mitigating spurious correlations and improving SSL robustness which are discussed in the related work section. We also clarified our experimental setup, including the use of multiple spurious correlations in the MIMIC-CXR dataset, and discussed the challenges of identifying spurious correlations on more complex data in the limitations section.

Please refer to the itemized responses for more details.

---

### Decision · Action_Editor_fcz5 · 2024-12-19

**Recommendation:** Accept with minor revision

**Comment:**

The reviewers were overall positive. They raised some concerns related to including more baselines, being clear about potential memory and compute implications of the technique, and providing more thorough consideration of the limitations of this approach and the evidence marshalled for it. The authors did a decent job of addressing the reviewers' concerns, and so, a decision of accept was reached. However, the authors need to provide a properly formatted, camera ready version, incoroporating all of the promised adjustments to the manuscript before final acceptance.

**Audience:**

This paper will be of interest to members of the TMLR community that are interested in imrpoving the robustness of SSL techniques.

**Claims And Evidence:**

This paper presents a technique for making self-supervised learning (SSL) less sensitive to spurious correlations. Specifically, the authors show that when spurious correlations exist in some data, the representations for data points that match the spurious correlations will converge faster than the representations that conflict with the spurious correlations. This leads the authors to their technique, which samples the data non-uniformly by giving more weight to samples that conflict with the spurious correlations, as determined by the speed at which the representations for those points change. The authors claim that this technique leads to pretrained representations that, when used for downstream classification, are more robust to spurious correlations and thus more accurate.

The authors present empirical results on multiple datasets, SSL losses, and network architectures to support their claims. Overall, the evidence is clear and convincing.